# On the Emergence of the Deviation from a Poisson Law in Stochastic Mathematical Models for Radiation-Induced DNA Damage: A System Size Expansion

**DOI:** 10.3390/e25091322

**Published:** 2023-09-11

**Authors:** Francesco Giuseppe Cordoni

**Affiliations:** Department of Civil, Environmental and Mechanical Engineering, University of Trento, 38123 Trento, Italy; francesco.cordoni@unitn.it

**Keywords:** biophysical modeling, radiation-induced DNA damage, system size expansion, DNA damage repair

## Abstract

In this paper, we study the system size expansion of a stochastic model for radiation-induced DNA damage kinetics and repair. In particular, we characterize both the macroscopic deterministic limit and the fluctuation around it. We further show that such fluctuations are Gaussian-distributed. In deriving such results, we provide further insights into the relationship between stochastic and deterministic mathematical models for radiation-induced DNA damage repair. Specifically, we demonstrate how the governing deterministic equations commonly employed in the field arise naturally within the stochastic framework as a macroscopic limit. Additionally, by examining the fluctuations around this macroscopic limit, we uncover deviations from a Poissonian behavior driven by interactions and clustering among DNA damages. Although such behaviors have been empirically observed, our derived results represent the first rigorous derivation that incorporates these deviations from a Poissonian distribution within a mathematical model, eliminating the need for specific ad hoc corrections.

## 1. Introduction

Radiotherapy is one of the most effective and used cancer treatment modalities [1]. Traditionally, radiotherapy relies on photons; however, in recent decades, there has been a growing interest in advanced radiotherapy using ion beams. Ion beams offer several advantages over photons [2], particularly their ability to release energy in a highly localized manner within tissues, potentially leading to a more effective biological response with reduced collateral effects in healthy tissues. Extensive research of the scientific community has focused on studying the effects of radiation on biological tissue, with DNA being identified as the most vulnerable target for radiation-induced damage leading to cell death [3]. Despite the theoretical advantages of using ion beams, further research is necessary to integrate this treatment modality into clinical practice fully. One significant challenge in the widespread adoption of ion beams lies in accurately determining their biological effects, as this is crucial for prescribing the most suitable treatment. Over the years, mathematical models have been developed to understand and predict the biological impact of ions on tissue, particularly in relation to DNA *Double Strand Breaks* (DSB) [4,5,6,7,8,9,10]. These mathematical approaches aim to describe the formation, progression, and clustering of DSBs, ultimately striving to predict the cell survival probability following radiation exposure.

Despite the inherently stochastic nature of biological pathways, most existing mathematical models, until now, have relied on deterministic frameworks with a priori assumptions about the Poisson distribution and disregarded the stochastic fluctuations in energy deposition. These fluctuations occur from cell to cell, particularly in complex radiation environments. In particular, the *Microdosimetric Kinetic Model* (MKM) [5,11], together with the *Local Effect Model* (LEM) [12,13], is one of the only two mechanistic models used in clinical treatment planning, which describes the temporal evolution of the average number of DNA damages in a single cell nucleus to obtain a linear–quadratic survival probability starting from energy deposition at the micron scale, in the order of the cell nucleus. A funding assumption of the linear–quadratic relation between dose and the corresponding survival probability is the Poissonian distribution for DNA damage. Evidence has emerged that in certain situations, such as for heavy ions or at high doses, the Poissonian assumption does not hold. Therefore, in recent years, the community tried to overcome this assumption with some *ad hoc* correction terms related to intercellular damage interaction [4,11,14,15,16,17]. Though attempts have been made to address the limitations of the Poissonian assumption, a stochastic representation that encompasses the spatial and temporal aspects of dose deposition was lacking in the description of radiation-induced DNA damage formation and dynamics.

Recently, a series of papers addressed these issues, and the *Generalized Stochastic Microdosimetric Model* (GSM${}^{2}$) [10,18,19,20,21] has been introduced. GSM${}^{2}\;$is a probabilistic model able to describe the time evolution of the DNA damage in a cell nucleus based on a differential equation governing the time evolution of the probability distribution of the number of DNA damages. Among the most relevant GSM${}^{2}\;$strengths, there is the capability to efficiently treat the several levels of spatiotemporal stochasticity happening during protracted irradiation without relying on the typically used Poissonian assumption on the number of DNA damages induced by radiation [19]. It is further described in [10,18] how different parameters, initial DNA damage distribution, or irradiation conditions can lead naturally to several possible probability distributions that can be significantly different from the typically assumed Poissonian law.

The main *master equation* governing the time evolution of the probability distribution of the number of DNA damages derived in [10] is non-linear due to the presence of a quadratic term that accounts for DNA clustering, which has been recognized as one of the main factors that leads to cell inactivation in radiobiology [7]. On the one side, this quadratic term plays a crucial role in the emergence of non-Poissonian behaviors; on the other side, it makes it difficult to obtain an explicit solution for the probability distribution of the number of DNA damages.

In this study, we present a system size expansion of the GSM${}^{2}\;$master equation based on the pioneering work [22]. The approximation that will be carried out in the present work is usually referred to in the literature as *system size expansion* [22,23], and it is widely used in the physics community to provide an appropriate macroscopic approximation of *microscopic* systems. As for all formal expansions, a suitable parameter is needed, around which the approximation is performed. In concrete applications, the domain size usually provides a suitable parameter to carry out a formal approximation, so that an asymptotic expansion of the main GSM${}^{2}\;$*master equation* will be carried out as the system size increases. It must be stressed that the approximation is generally valid as the number of lesions increases [23], so that the approximation derived in the present work provides a relevant description for high-dose irradiation, where the number of lesions even in small domains is high. Such a case is of particular relevance, since most of the existing models fail to give a precise description of the cell survival probability at high doses [4], and suitable correction terms are needed to match experimental data.

We will derive an asymptotic expansion for the GSM${}^{2}\;$master equation computing both the macroscopic limit and the fluctuation around such a macroscopic limit. Besides allowing us to calculate an approximate distribution for the number of DNA damage, the expansion derived in the present work provides further insights into the relationship between stochastic and deterministic mathematical models, already highlighted in previous works [18,19]. Having in mind the above-mentioned regimes of validity for the proposed expansion, we will further strengthen the connection between GSM${}^{2}\;$and the MKM, showing that as the system size increases, the *master equation* derived in [10] converges toward the main deterministic kinetic equations of the MKM [4,5]. This emphasizes how deterministic macroscopic behavior emerges from stochastic microscopic fluctuations. We will go a step further so that we will also prove a suitable central limit theorem, in the sense that we will characterize the stochastic fluctuations around the macroscopic average value. These fluctuations are usually assumed to be Poissonian. By contrast, we will show that such fluctuation described by the macroscopic approximation of GSM${}^{2}\;$are Gaussian-distributed, as described by a *linear Fokker–Planck equation* (FPE) [23]. Recalling that for large mean value λ, a Poisson random variable of mean λ is approximated in a probabilistic sense by a Gaussian random variable with mean and variance λ, we will show that the derived limiting model can be seen as a correction around a Poisson distribution due to the clustering of lesions. In this sense, the present work shows how typical non-Poissonian correction terms of the MKM that have been proposed over the years naturally emerge in the fully probabilistic description of the GSM${}^{2}$. Lastly, it is worth stressing that the present paper sheds light on another possible future connection to existing radiobiological models. In fact, in the literature, some models have been derived that attempt to describe DNA damage at high doses using a Gaussian formulation of a multi-hit model (MHM) [24,25]. It has already been shown in [19] that GSM${}^{2}\;$is closely connected to some multi-hit models [26,27], so the results derived in the present paper further connect GSM${}^{2}\;$to the multi-hit models derived in the literature.

The main contributions of the present paper are:(i)to derive a system size expansion for the master equation governing GSM${}^{2}\;$studying both the macroscopic limit and the fluctuations around the average;(ii)to show how the nonlinear terms accounting for DNA clustering give rise to a non-Poissonian behavior;(iii)to shed light on another insightful connection between existing radiobiological models.

The present work is structured as follows: Section 2.1 recalls the basic facts of GSM${}^{2}$. Section 3.1 contains the main result of the present research, with the formal asymptotic expansion and a rigorous description of the stochastic fluctuations around the macroscopic average value. Section 4 provides some numerical results on the derived approximation.

## 2. Material and Methods

### 2.1. The Generalized Stochastic Microdosimetric Model and the Microdosimetric Kinetic Model

GSM${}^{2}\;$ [10] is a stochastic model that provides a probabilistic description of DNA damage formation and evolution, with particular attention to the link to DNA damage formation and energy deposition. The final goal of the model is to overcome existing models mainly based on the Poissonian assumption of energy deposition to provide a better characterization of some relevant biological endpoints such as the cell survival fraction.

GSM${}^{2}$ considers two types of DNA damage, called sub-lethal and lethal lesions. Lethal lesions *Y* represent damage that cannot be repaired, leading to cell inactivation. By contrast, sublethal lesions *X* can be repaired at rate *r* or become a lethal lesion either by direct death at rate *a* or interacting with another sublethal lesion at rate *b*. This latter term *b* accounts for the clustering of DNA damage and gives rise to a nonlinearity in the governing master equation.

Denoted by $\left(Y(t),X(t)\right)$ is the state of the system at time *t*, where *X* and *Y* are two N−valued random variables counting the number of the lethal and sub-lethal lesion, respectively. In the following, we will consider a standard complete filtered probability space $\left(\Omega ,F,{\left(F_{t}\right)}_{t\geq 0},P\right)$ satisfying usual assumptions, namely right-continuity and saturation by P-null sets.

The above reasoning can be represented by the following pathways:$$\begin{array}{cc} & X\overset{r}{\to }\emptyset ,\\ & X\overset{a}{\to }Y,\\ & X+X\overset{b}{\to }Y, \end{array}$$
for three positive constant rates, *r*, *a*, and *b*.

In what follows, we denote by:$$\begin{array}{cc} & p_{t_{0},y_{0},x_{0}}\left(t,y,x\right):=p\left(t,y,x|t_{0},y_{0},x_{0}\right)=P\left(\left.\left(Y(t),X(t)\right)=\left(y,x\right)\right|\left(Y\left(t_{0}\right),X\left(t_{0}\right)\right)=\left(y_{0},x_{0}\right)\right). \end{array}$$
If no confusion is possible, we will avoid stating the initial time and state, writing for short p(t,y,x).

Following [10], the *microdosimetric master equation* (MME) can be derived:(1)$$\begin{array}{cc} \frac{\partial }{\partial t}p\left(t,y,x\right) & =-\left[(a+r)x+bx(x-1)\right]p\left(t,y,x\right)+\left(x+1\right)rp\left(t,y,x+1\right)+\\ \; & +(x+1)ap(t,y-1,x+1)+(x+2)(x+1)bp(t,y-1,x+2). \end{array}$$

The MME (Equation 1) can be written for short as:(2)$$\begin{array}{cc} \frac{\partial }{\partial t}p\left(t,y,x\right) & =\left(E^{-1,2}-1\right)\left[x(x-1)bp(t,y,x)\right]+\left(E^{-1,1}-1\right)\left[xap(t,y,x)\right]+\\ \; & +\left(E^{0,1}-1\right)\left[xrp(t,y,x)\right]=\\ \; & =E^{-1,2}\left[x(x-1)bp(t,y,x)\right]+E^{-1,1}\left[xap(t,y,x)\right]+E^{0,1}\left[xrp(t,y,x)\right], \end{array}$$
where above we have denoted the creation operators as:$$E^{i,j}\left[f(y,x)\right]:=\left(E^{i,j}-1\right)\left[f(y,x)\right]:=f\left(y+i,x+j\right)-f\left(y,x\right).$$

The above MME (Equation 2) is coupled with an initial distribution:$$p_{0}\left(y,x\right):=p\left(0,y,x\right),$$
as described in [10,18].

**Remark** **1.**
*The above choice is made to closely follow the existing literature on the topic. However, other choices for the pathways r, a, and b can be made. For instance, it could be assumed that the death rate a is logistic, including an increment in the death as the number of lesions becomes bigger. In such a case, the MME would become:*

(3)
$$\begin{array}{cc} \frac{\partial }{\partial t}p\left(t,y,x\right) & =E^{-1,2}\left[x(x-1)bp(t,y,x)\right]+E^{-1,1}\left[x\left(a+\bar{a}x\right)p\left(t,y,x\right)\right]+E^{0,1}\left[xrp(t,y,x)\right]. \end{array}$$



It has been shown in [10] that GSM${}^{2}\;$is closely connected to the MKM, where, in fact, the former represents a stochastic reformulation of the latter. The MKM postulates the same assumptions of GSM${}^{2}\;$with two key additions. First, the MKM considers the time evolution for the average number of lethal lesions $\bar{y}$ and sublethal lesions $\bar{x}$, and second, $\bar{y}$ is assumed to be Poissonian-distributed.

In particular, the MKM assumes the $\bar{y}$ and $\bar{x}$ follows the set of coupled ODE:(4)$$\left\{\begin{array}{c} \frac{d}{dt}\bar{y}\left(t\right)=a\bar{x}\left(t\right)+b{\bar{x}}^{2}\left(t\right),\\ \frac{d}{dt}\bar{x}\left(t\right)=-\left(a+r\right)\bar{x}\left(t\right)-2b{\bar{x}}^{2}\left(t\right). \end{array}\right.$$

Typically, it is further assumed that $\left(a+r\right)x>>2bx^{2}$, so that Equation (Equation 4) reduces to:(5)$$\left\{\begin{array}{c} \frac{d}{dt}\bar{y}\left(t\right)=a\bar{x}\left(t\right)+b{\bar{x}}^{2}\left(t\right),\\ \frac{d}{dt}\bar{x}\left(t\right)=-\left(a+r\right)\bar{x}\left(t\right). \end{array}\right.$$

The connection between the MME (Equation 2) and the system of ODE (Equation 4) has already been shown in [10,18], and this connection will be further deepened in the present paper.

## 3. Theory and Calculations

### 3.1. Macroscopic Description for the GSM${}^{2}$

In the present Section, we will derive a rigorous expansion that provides a macroscopic linear approximation counter-part of the master equation derived in [10]. The following expansion is at the basis of a macroscopic deterministic description of a microscopic stochastic system.

In the following, we will assume that the coefficients of the master Equation (Equation 2) depend on a parameter K in a suitable manner; namely, they are of order O(1) with respect to the parameter K; under this assumption, we are able to characterize the limit for the master Equation (Equation 2) as K→∞. As mentioned in the introduction, the typical approach is to consider K as the system size, from which the name *system size expansion* is derived.

We will prove both the convergence of the microscopic system towards a macroscopic mean value, which corresponds to *the law of large numbers*, and also provide a description for the fluctuations of the system around such a mean value. This description of the fluctuations allows us to describe the system in terms of an FPE so that we will show that the order of the fluctuation is not Poissonian, as typically assumed in most of the existing literature on the subject. In this sense, the current research highlights how GSM${}^{2}\;$provides a rigorous non-Poissonian correction to the MKM.

In the following, in order to carry out the expansion, we assume that the parameter *b* depends on the K as ${\tilde{b}}_{K}:=\frac{b}{K}$. Therefore, the MME (Equation 2) now becomes:(6)$$\begin{array}{cc} \frac{\partial }{\partial t}p\left(t,y,x\right) & =E^{-1,2}\left[x\left(x-1\right){\tilde{b}}_{K}p\left(t,y,x\right)\right]+E^{-1,1}\left[xap(t,y,x)\right]+E^{0,1}\left[xrp(t,y,x)\right]. \end{array}$$

The main idea that will be carried out in the current section is to let K→∞ in order to approximate the master equation by a continuous equation. The first-order approximation will satisfy a linear FPE, whose marginals, under some specific initial distribution to be better specified in later sections, can be shown to be Gaussian-distributed. The derived Gaussian approximation for the lesion distribution will be shown to provide better insights than the classical Poissonian hypothesis regarding lethal damage.

In order to prove the expansion, we set:(7)$$\begin{array}{cc} X(t) & =K\bar{x}\left(t\right)+\sqrt{K}\xi \left(t\right),\\ Y(t) & =K\bar{y}\left(t\right)+\sqrt{K}\upsilon \left(t\right), \end{array}$$
with ${\left(\xi (t)\right)}_{t\geq 0}$ and ${\left(\upsilon (t)\right)}_{t\geq 0}$ being two stochastic processes, so that for any t≥0, ξ(t) and υ(t) are two centered random variables, i.e., with null mean value, whereas $\bar{x}\left(t\right)$ and $\bar{y}\left(t\right)$ are two suitable deterministic functions to be derived later. Heuristically speaking, $\bar{x}$ and $\bar{y}$ will play the role of the macroscopic deterministic behavior, which we will show to agree with the differential equations governing the MKM, as given in (Equation 4). Therefore, the above assumptions can be interpreted intuitively as an expansion of variables *x* and *y* around the macroscopic behavior, whereas the terms ξ and υ represent the fluctuations around the mean value.

It is worth noting that the following holds true:(8)$$\begin{array}{cc} E[X(t)] & =K\bar{x}\left(t\right)+\sqrt{K}E\left[\xi \left(t\right)\right],\\ E[Y(t)] & =K\bar{y}\left(t\right)+\sqrt{K}E\left[\upsilon \left(t\right)\right],\\ Var[X(t)] & =KVar[\xi (t)],\\ Var[Y(t)] & =KVar[\upsilon (t)]\;. \end{array}$$

Define the new distribution with respect to the new variables as:$$p\left(t,y,x\right)=p\left(t,K\bar{x}+\sqrt{K}\xi ,K\bar{y}+\sqrt{K}\upsilon \right)=P\left(t,\upsilon ,\xi \right)\;.$$

The standard chain rule applied to P(t,υ,ξ) yields:$$\frac{\partial }{\partial t}p\left(t,y,x\right)=\frac{\partial }{\partial t}P\left(t,\upsilon ,\xi \right)\left(1+\frac{\partial }{\partial t}\upsilon +\frac{\partial }{\partial t}\xi \right)\;,$$
so that inverting transformation (Equation 7) for υ and ξ gives:$$\frac{\partial }{\partial t}p\left(t,y,x\right)=\frac{\partial }{\partial t}P\left(t,\upsilon ,\xi \right)-\sqrt{K}\frac{d}{dt}\bar{y}\frac{\partial }{\partial \upsilon }P\left(t,\upsilon ,\xi \right)-\sqrt{K}\frac{d}{dt}\bar{x}\frac{\partial }{\partial \xi }P\left(t,\upsilon ,\xi \right)\;.$$

Regarding the step operators appearing in the MME (Equation 6), it can be shown that the following holds true:$$E^{i,j}=\frac{1}{\sqrt{K}}\left(i\frac{\partial }{\partial \upsilon }+j\frac{\partial }{\partial \xi }\right)+\frac{1}{2}\frac{1}{K}{\left(i\frac{\partial }{\partial \upsilon }+j\frac{\partial }{\partial \xi }\right)}^{2}\;.$$

The above computations substituted into Equation (Equation 6) yields:(9)$$\begin{array}{cc} \frac{\partial }{\partial t}P\left(t,\upsilon ,\xi \right) & =\sqrt{K}\frac{d}{dt}\bar{y}\frac{\partial }{\partial_{\upsilon }}P\left(t,\upsilon ,\xi \right)+\sqrt{K}\frac{d}{dt}\bar{x}\frac{\partial }{\partial_{\xi }}P\left(t,\upsilon ,\xi \right)+\\ \; & +\frac{b}{K}\left[\frac{1}{\sqrt{K}}\left(2\frac{\partial }{\partial_{\xi }}-\frac{\partial }{\partial_{\upsilon }}\right)\right]\left(K\bar{x}+\sqrt{K}\xi \right)\left(K\bar{x}+\sqrt{K}\xi -1\right)P\left(t,\upsilon ,\xi \right)+\\ \; & +\frac{b}{K}\left[\frac{1}{2}\frac{1}{K}{\left(2\frac{\partial }{\partial_{\xi }}-\frac{\partial }{\partial_{\upsilon }}\right)}^{2}\right]\left(K\bar{x}+\sqrt{K}\xi \right)\left(K\bar{x}+\sqrt{K}\xi -1\right)P\left(t,\upsilon ,\xi \right)+\\ \; & +a\left[\frac{1}{\sqrt{K}}\left(\frac{\partial }{\partial_{\xi }}-\frac{\partial }{\partial_{\upsilon }}\right)+\frac{1}{2}\frac{1}{K}{\left(\frac{\partial }{\partial_{\xi }}-\frac{\partial }{\partial_{\upsilon }}\right)}^{2}\right]\left(K\bar{x}+\sqrt{K}\xi \right)P\left(t,\upsilon ,\xi \right)+\\ \; & +r\left[\frac{1}{\sqrt{K}}\frac{\partial }{\partial_{\xi }}+\frac{1}{2}\frac{1}{K}\frac{\partial^{2}}{\partial_{\xi }^{2}}\right]\left(K\bar{x}+\sqrt{K}\xi \right)P\left(t,\upsilon ,\xi \right)\;. \end{array}$$

Grouping the terms of order $\sqrt{K}$, we obtain:(10)$$\begin{array}{cc} ∼\sqrt{K} & :\;\frac{d}{dt}\bar{y}\frac{\partial }{\partial_{\upsilon }}P\left(t,\upsilon ,\xi \right)+\frac{d}{dt}\bar{x}\frac{\partial }{\partial_{\xi }}P\left(t,\upsilon ,\xi \right)+a\left(\frac{\partial }{\partial_{\xi }}-\frac{\partial }{\partial_{\upsilon }}\right)\bar{x}P\left(t,\upsilon ,\xi \right)+\\ \; & +r\frac{\partial }{\partial_{\xi }}\bar{x}P\left(t,\upsilon ,\xi \right)+2b{\bar{x}}^{2}\frac{\partial }{\partial_{\xi }}P\left(t,\upsilon ,\xi \right)-b{\bar{x}}^{2}\frac{\partial }{\partial_{\upsilon }}P\left(t,\upsilon ,\xi \right)\;. \end{array}$$

In order to compensate for the terms of order $\sqrt{K}$, we set the macroscopic system as:(11)$$\left\{\begin{array}{c} \frac{d}{dt}\bar{y}=a\bar{x}+b{\bar{x}}^{2}\;\\ \frac{d}{dt}\bar{x}=-\left(a+r\right)\bar{x}-2b{\bar{x}}^{2}\;, \end{array}\right.$$
so that all terms or order $\sqrt{K}$ in Equation (Equation 10) vanishes. Therefore, we have shown that the macroscopic limit of GSM${}^{2}\;$MME coincides with the main deterministic governing equation of the MKM (Equation 4).

Explicit solutions to system (Equation 11) can be derived with the further property that they are globally stable and converging to a stationary solution [19]. Consider first:(12)$$\frac{d}{dt}\bar{x}\left(t\right)=-\left(a+r\right)\bar{x}\left(t\right)-2b{\bar{x}}^{2}\left(t\right)\;,\;\bar{x}\left(0\right)=x_{0}\;.$$

Such an Equation (Equation 12) is known as the Bernoulli equation. Applying the transformation $u=\frac{1}{\bar{x}}$ leads to the following differential equation:$$\frac{d}{dt}u\left(t\right)=\left(a+r\right)u\left(t\right)+2b\;.$$
This last equation is a linear equation in *u*, so the explicit solution is given by:$$u\left(t\right)=ce^{(a+r)t}-\frac{2b}{a+r}\;.$$
Coming back to the original equation, we obtain:$$\bar{x}\left(t\right)=\frac{a+r}{ce^{(a+r)t}-2b}\;,$$
with
$$c:=\frac{a+r}{x_{0}}+2b\;.$$

We can, therefore, substitute $\bar{x}$ into Equation (Equation 11) to obtain:$$\begin{array}{cc} \bar{y}\left(t\right) & =y_{0}+\frac{a+r}{4b-2ce^{t(a+r)}}+\frac{r}{2b}\left(\left(a+r\right)t-\log\left[\left|2b-ce^{t(a+r)}\right|\right]\right)\;. \end{array}$$

We can eventually calculate the long-time convergence toward the stationary solution of the above equations to be:$$\begin{array}{cc} \lim_{t\to \infty }\bar{x}\left(t\right) & =:{\bar{x}}_{\infty }=0\;,\\ \lim_{t\to \infty }\bar{y}\left(t\right) & =:{\bar{y}}_{\infty }=y_{0}-\frac{r}{2b}\log\left(\frac{a+r}{x_{0}}+2b\right)\;. \end{array}$$

**Remark** **2.**
*It is worth remarking that, for low-dose and sparsely ionizing radiation, such as X-rays or high-energy protons, the following assumption typically holds true, $\left(a+r\right)x>>2bx^{2}$; therefore, the above calculations simplify so that the explicit solution to Equation (Equation 11) is given by [5,28]:*

(13)
$$\left\{\begin{array}{c} \bar{y}\left(t\right)=y_{0}+ax_{0}\left(\frac{1-e^{-(a+r)t}}{a+r}\right)+bx_{0}^{2}\left(\frac{1-e^{-2(a+r)t}}{a+r}\right)\;\\ \bar{x}\left(t\right)=x_{0}e^{-(a+r)t}\;. \end{array}\right.$$


*In particular, Equation (Equation 13) converges as t→∞ towards:*

$$\lim_{t\to \infty }\bar{y}\left(t\right)=y_{0}+x_{0}\left(\frac{a}{a+r}+x_{0}\frac{b}{a+r}\right)\;,\;\lim_{t\to \infty }\bar{x}\left(t\right)=0\;.$$



### 3.2. The Linear Noise Approximation and Moments Estimates

Having cancelled out terms of order $\sqrt{K}$, taking the limit as K→∞, all terms containing K vanish and only the zero−th order terms remain, yielding:(14)$$\begin{array}{cc} \frac{\partial }{\partial t}P\left(t,\upsilon ,\xi \right) & =2b\left(2\frac{\partial }{\partial_{\xi }}-\frac{\partial }{\partial_{\upsilon }}\right)\left[\bar{x}\xi P\left(t,\upsilon ,\xi \right)\right]+\frac{1}{2}b{\left(2\frac{\partial }{\partial_{\xi }}-\frac{\partial }{\partial_{\upsilon }}\right)}^{2}\left[{\bar{x}}^{2}P\left(t,\upsilon ,\xi \right)\right]+\\ \; & +a\left(\frac{\partial }{\partial_{\xi }}-\frac{\partial }{\partial_{\upsilon }}\right)\left[\xi P(t,\upsilon ,\xi )\right]+r\frac{\partial }{\partial_{\xi }}\left[\xi P(t,\upsilon ,\xi )\right]+\\ \; & +\frac{1}{2}a{\left(\frac{\partial }{\partial_{\xi }}-\frac{\partial }{\partial_{\upsilon }}\right)}^{2}\left[\bar{x}P\left(t,\upsilon ,\xi \right)\right]+\frac{1}{2}r\frac{\partial^{2}}{\partial_{\xi }^{2}}\left[\bar{x}P\left(t,\upsilon ,\xi \right)\right]+\\ \; & =\frac{\partial }{\partial_{\xi }}\left[\xi P(t,\upsilon ,\xi )\right]\left(4b\bar{x}+a+r\right)-\frac{\partial }{\partial_{\upsilon }}\left[\xi P(t,\upsilon ,\xi )\right]\left(2b\bar{x}+a\right)+\\ \; & -\partial_{\xi \upsilon }P\left(t,\upsilon ,\xi \right)\left(2b{\bar{x}}^{2}+a\bar{x}\right)+\\ \; & +\frac{1}{2}\frac{\partial^{2}}{\partial_{\xi }^{2}}P\left(t,\upsilon ,\xi \right)\left(\left(a+r\right)\bar{x}+4b{\bar{x}}^{2}\right)+\frac{1}{2}\frac{\partial^{2}}{\partial_{\upsilon }^{2}}P\left(t,\upsilon ,\xi \right)\left(a\bar{x}+b{\bar{x}}^{2}\right)\;. \end{array}$$

Equation (Equation 14) is a linear *Fokker–Planck* equation of dimension 2 that describes the fluctuations of the system around the average values $\bar{x}\left(t\right)$ and $\bar{y}\left(t\right)$. The solution to the linear FP Equation (Equation 14), under suitable initial conditions that will be specified later, can be shown to be the bi-dimensional Gaussian density.

Until now, we have avoided explicitly considering the initial condition both for the original MME (Equation 6) and for the approximating linear FPE (Equation 14).

As shown in [18], much of the stochasticity regarding lesion formation lies in the initial condition, in the sense that the distribution of initial lethal and sub-lethal damage deeply affects the subsequent time evolution of the probability density function. We will avoid an extensive treatment of such a topic in the present paper and focus more on the stochasticities inherent to the kinetics of the interaction of DNA damages, considering instead two simple and yet relevant cases for the initial damage distribution.

Let us start by assuming that the initial number of lesions is deterministic and is given by $\left(y_{0},x_{0}\right)$. We, therefore, equip the MME (Equation 6) with a deterministic initial condition given by:$$p\left(0,y,x\right)=\delta \left(x-x_{0}\right)\delta \left(y-y_{0}\right)\;,$$with $\delta (x-x_{0})$ and $\delta (y-y_{0})$ being the Dirac delta centered at $x_{0}$ and $y_{0}$, respectively. It can be shown that [23,29] the solution to the linear FPE (Equation 14) is given by a bivariate Gaussian distribution:$$P\left(t,\upsilon ,\xi \right)=\frac{1}{2\pi }\frac{1}{\sqrt{detC}}\exp\left\{-\frac{1}{2}{\left(\upsilon -\bar{\upsilon },\xi -\bar{\xi }\right)}^{T}C^{-1}\left(\upsilon -\bar{\upsilon },\xi -\bar{\xi }\right)\right\}\;,$$
where $\bar{\upsilon }$ and $\bar{\xi }$ are the mean values and *C* is the covariance matrix with entries:$$C=\left(\begin{array}{ccc} c_{\upsilon \upsilon } & c_{\xi \upsilon }\\ c_{\xi \upsilon } & c_{\xi \xi } & \end{array}\right)\;,$$
where $c_{\upsilon \upsilon }$, resp. $c_{\xi \upsilon }$, and resp. $c_{\xi \xi }$ are the variance of υ, resp. covariance of ξ and υ, and resp. variance of ξ. It is worth stressing that, given the properties of the multivariate Gaussian distributions, ξ and υ are univariate Gaussian random variables.

Upon the multiplication of Equation (Equation 14) by ξ and υ, it follows after integrating by parts that the first moment of ξ and υ satisfies:(15)$$\begin{array}{cc} \frac{d}{dt}\bar{\upsilon } & =\left(2b\bar{x}+a\right)\bar{\xi }\;,\;\bar{\upsilon }\left(0\right)=0\;,\\ \frac{d}{dt}\bar{\xi } & =-\left(4b\bar{x}+a+r\right)\bar{\xi }\;,\;\bar{\xi }\left(0\right)=0\;. \end{array}$$
It immediately follows from Equation (Equation 15) that:$$\bar{\xi }\left(t\right)=\bar{\upsilon }\left(t\right)\equiv 0\;.$$
This result is in agreement with the fact that ξ and υ are centered random variables.

Multiplying Equation (Equation 14) by $\xi^{2}$, ξυ, and $\upsilon^{2}$, we obtain, again after integration by parts, that the variance and covariance satisfy the following set of coupled ODEs:(16)$$\left\{\begin{array}{cc} \frac{d}{dt}c_{\upsilon \upsilon } & =2\left(2b\bar{x}+a\right)c_{\xi \upsilon }+a\bar{x}+b{\bar{x}}^{2}\;,\\ \frac{d}{dt}c_{\xi \upsilon } & =\left(2b\bar{x}+a\right)c_{\xi \xi }-\left(4b\bar{x}+a+r\right)c_{\xi \upsilon }-\left(2b{\bar{x}}^{2}+a\bar{x}\right)\;,\\ \frac{d}{dt}c_{\xi \xi } & =-2\left(4b\bar{x}+a+r\right)c_{\xi \xi }+\left(a+r\right)\bar{x}+4b{\bar{x}}^{2}\;, \end{array}\right.$$
with the initial condition $c_{\upsilon \upsilon }\left(0\right)=c_{\xi \upsilon }\left(0\right)=c_{\xi \xi }\left(0\right)=0$. The last two equations in (Equation 16) can be computed to be:(17)$$\left\{\begin{array}{cc} c_{\xi \xi }\left(t\right) & =\frac{e^{2t(a+r)}\left(\left(a+r\right)\left(-4b^{2}e^{t(-a-r)}-4bct\left(a+r\right)+c^{2}e^{t(a+r)}\right)+c\left(a+r\right)\left(4b^{2}-c^{2}\right)+ac\right)}{{\left(ce^{t(a+r)}-2b\right)}^{4}}\;,\\ c_{\xi \upsilon }\left(t\right) & =\frac{e^{t(a+r)}\left(-\frac{ce^{t(a+r)}\left(2a^{2}\left(2brt+b\right)+ar\left(-4b\left(b-2rt-1\right)+c^{2}-1\right)+r^{2}\left(2b\left(-2b+2rt+1\right)+c^{2}\right)\right)}{2b\left(a+r\right)\left(ce^{t(a+r)}-2b\right)}\right)}{{\left(ce^{t(a+r)}-2b\right)}^{2}}+\\ \; & +\frac{e^{t(a+r)}\left(\frac{c^{2}e^{2t(a+r)}\left(-4ab^{2}+4bt{\left(a+r\right)}^{2}+ac^{2}-a-4b^{2}r+c^{2}r\right)}{4b{\left(ce^{t(a+r)}-2b\right)}^{2}}+2bre^{t(-a-r)}-t\left(a+r\right)\left(ac-r\right)+c_{\xi \upsilon }\right)}{{\left(ce^{t(a+r)}-2b\right)}^{2}}\;, \end{array}\right.$$
with $c_{\xi \upsilon }$ a suitable constant to ensure the initial condition. It can be seen that it holds:$$\lim_{t\to \infty }c_{\xi \xi }\left(t\right)=\lim_{t\to \infty }c_{\xi \upsilon }\left(t\right)=0\;.$$

In particular, we are mostly concerned with the term $c_{\upsilon \upsilon }$ and with its stationary solution, as we aim to show that the distribution of lethal lesions differs from a Poisson distribution, as it is typically assumed in radiobiological models. It can, thus, be noticed that, integrating the third equation in (Equation 16), we obtain:(18)$$\begin{array}{cc} c_{\upsilon \upsilon }\left(t\right) & =\int_{0}^{t}\left[2\left(2b\bar{x}\left(s\right)+a\right)c_{\xi \upsilon }\left(s\right)+a\bar{x}\left(s\right)+b{\bar{x}}^{2}\left(s\right)\right]ds=\\ \; & =\bar{y}\left(t\right)+\int_{0}^{t}2\left(2b\bar{x}\left(s\right)+a\right)c_{\xi \upsilon }\left(s\right)ds=\bar{y}\left(t\right)-\delta \left(t\right)\;, \end{array}$$
with $\bar{y}\left(t\right)$ being the mean value for lethal lesions, as computed in Equation (Equation 11), and:$$\delta \left(t\right):=-\int_{0}^{t}2\left(2b\bar{x}\left(s\right)+a\right)c_{\xi \upsilon }\left(s\right)ds\;.$$
The negative sign in δ is used to emphasize that the covariance is, in fact, negative, since a decrease in sublethal lesions correlates with an increase in lethal lesions.

The long time behaviour for $c_{\upsilon \upsilon }$ can be explicitly computed using Equation (Equation 18) to be:(19)$$\begin{array}{cc} \lim_{t\to \infty }c_{\upsilon \upsilon }\left(t\right) & =:{\bar{y}}_{\infty }-\delta_{\infty }\;, \end{array}$$
with ${\bar{y}}_{\infty }$ being the long-time solution to the macroscopic average value $\bar{y}\left(t\right)$.

Recalling that for a large mean value, a Poisson distribution can be approximated by a Gaussian random variable with equal mean and variance, in order to infer that the lethal lesion distribution obeys a Poisson random variable, we must obtain $\lim_{t\to \infty }c_{\upsilon \upsilon }\left(t\right)={\bar{y}}_{\infty }$. By contrast, the above calculations show that the variance is given by the average value corrected by a term given by the covariance of two types of lesions. In particular, as there is a negative correlation between the two variables, we can infer that the lethal lesion distribution is almost a Poisson random variable, where the variance is adjusted by subtracting a term due to pairwise interactions.

#### Moments Estimates for a Stochastic Initial Condition

In general, we cannot expect the initial number of lesions to be deterministic, so previous arguments must be slightly modified.

To explicitly compute the marginal distribution for the solution to the linear Fokker–Planck Equation (Equation 14), we assume the initial distribution for the MME to be normally distributed with mean $\left(y_{0},x_{0}\right)$ and variance Σ. It is worth remarking that such an assumption is not restrictive, as the standard assumption for the initial condition is to be a Poisson random variable, which, as mentioned above, under certain assumptions, can be approximated by a Gaussian random variable.

In particular, we assume that the initial number of lethal and sublethal lesions follows a Gaussian random variable with mean and variance given by $x_{0}$ and $y_{0}$:(20)$$p\left(0,y,x\right)=\frac{1}{2\pi }\frac{1}{\sqrt{det\Sigma }}\exp\left\{-\frac{1}{2}{\left(y-y_{0},x-x_{0}\right)}^{T}\Sigma^{-1}\left(y-y_{0},x-x_{0}\right)\right\}\;,$$
with
$$\Sigma =\left(\begin{array}{cc} x_{0} & 0\\ 0 & y_{0} \end{array}\right)\;.$$

Similar arguments as above imply that the initial condition for the linear FPE (Equation 14), under Equation (Equation 20), becomes a centered Gaussian random variable:$$P\left(0,\upsilon ,\xi \right)=\frac{1}{2\pi }\frac{1}{\sqrt{det\Sigma }}\exp\left\{-\frac{1}{2}{\left(\upsilon ,\xi \right)}^{T}\Sigma^{-1}\left(\upsilon ,\xi \right)\right\}\;.$$
Therefore, the initial fluctuations around the mean value are Gaussian-distributed with a null average.

Therefore, all calculations above follow alike, implying that, again, the solution to the linear Fokker-Planck Equation (Equation 14) is given by:$$P\left(t,\upsilon ,\xi \right)=\frac{1}{2\pi }\frac{1}{\sqrt{detC}}\exp\left\{-\frac{1}{2}{\left(\upsilon ,\xi \right)}^{T}C^{-1}\left(\upsilon ,\xi \right)\right\}\;,$$
where now the covariance matrix incorporates the initial stochastic condition so that its entries satisfy the following set of differential equations:(21)$$\left\{\begin{array}{cc} \frac{\partial }{\partial t}c_{\upsilon \upsilon } & =2\left(2b\bar{x}+a\right)c_{\xi \upsilon }+a\bar{x}+b{\bar{x}}^{2}\;,\;c_{\upsilon \upsilon }\left(0\right)=y_{0}\;,\\ \frac{\partial }{\partial t}c_{\xi \upsilon } & =\left(2b\bar{x}+a\right)c_{\xi \xi }-\left(4b\bar{x}+a+r\right)c_{\xi \upsilon }-2\left(4b{\bar{x}}^{2}+2a\bar{x}\right)\;,\;c_{\xi \upsilon }\left(0\right)=0\;,\\ \frac{\partial }{\partial t}c_{\xi \xi } & =-2\left(4b\bar{x}+a+r\right)c_{\xi \xi }+\left(a+r\right)\bar{x}+2b{\bar{x}}^{2}\;,\;c_{\xi \xi }\left(0\right)=x_{0}\;. \end{array}\right.$$

Analogously to what is shown at the end of Section 3.2, the variance of lethal lesion obeys:$$\begin{array}{c} c_{\upsilon \upsilon }\left(t\right)=\bar{y}\left(t\right)-\delta \left(t\right)\;, \end{array}$$
with $\bar{y}\left(t\right)$ the average deterministic value and:$$\delta \left(t\right)=-\int_{0}^{t}2\left(2b\bar{x}\left(s\right)+a\right)c_{\xi \upsilon }\left(s\right)ds\;,$$
so that, again, the variance for the lethal lesion is given by the macroscopic mean corrected by a covariance term.

**Remark** **3.**
*The solution to the linear FPE (Equation 14) can be shown [29] to be the probability density function of the time-dependent Ornstein–Uhlenbeck (OU) process, defined as:*

(22)
$$dZ\left(t\right)=-A\left(t\right)Z\left(t\right)dt+Q\left(t\right)dW\left(t\right)\;,\;\begin{array}{c} Z\left(0\right)=z_{0} \end{array}\;,$$

*with W a bidimensional standard Brownian motion, Z=(υ,ξ), and $z_{0}$ a bivariate Gaussian random variable with mean $\left(x_{0},y_{0}\right)$ and variance:*

$$\Sigma =\left(\begin{array}{cc} x_{0} & 0\\ 0 & y_{0} \end{array}\right)\;.$$


*Additionally:*

$$\begin{array}{cc} A(t) & =\left(\begin{array}{cc} 0 & -2b\bar{x}\left(t\right)-a\\ 0 & 4b\bar{x}\left(t\right)+a+r \end{array}\right)\;,\\ Q(t) & =\left(\begin{array}{cc} \sqrt{a\bar{x}\left(t\right)+b{\bar{x}}^{2}\left(t\right)+\frac{{\left(a\bar{x}\left(t\right)+b{\bar{x}}^{2}\left(t\right)\right)}^{2}}{\left(a+r\right)\bar{x}\left(t\right)+4b{\bar{x}}^{2}\left(t\right)}} & -\frac{a\bar{x}\left(t\right)+b{\bar{x}}^{2}\left(t\right)}{\sqrt{\left(a+r\right)\bar{x}\left(t\right)+4b{\bar{x}}^{2}\left(t\right)}}\\ 0 & \sqrt{\left(a+r\right)\bar{x}\left(t\right)+4b{\bar{x}}^{2}\left(t\right)}\;. \end{array}\right) \end{array}$$


*By simulating various trajectories of the OU process described in Equation (Equation 22), we can effectively estimate the solution to the FPE given by Equation (Equation 14). Furthermore, it is important to note that the boundary x=0 acts as an absorbing boundary in the original GSM${}^{2}\;$. This implies that once the number of sublethal lesions reaches zero, it remains at zero. Therefore, to guarantee the positivity of the solution to the FPE defined in Equation (Equation 14), it is necessary to impose a similar boundary condition on the OU process in Equation (Equation 22). This ensures that the number of lethal lesions remains positive and is absorbed at zero upon reaching the boundary.*


**Remark** **4.**
*It has been shown in [23,30] that a different and yet related Fokker–Planck equation can be obtained without any truncation at first order. In fact, if the above assumption on b holds true, then the master Equation (Equation 2), following [23] (Chapter 7.5), can be expanded as:*

$$\frac{\partial }{\partial t}p\left(t,y,x\right)=-\sum_{n}\sum_{\left(i,j\right)}\frac{{\left((i,j)\cdot \nabla \right)}^{n}}{n!}\left[C^{\left(i,j\right)}p\left(t,y,x\right)\right]\;,$$

*for a suitable term $C^{\left(i,j\right)}$.*

*Truncating at the second order, we obtain the following Fokker–Planck equation:*

(23)
$$\begin{array}{cc} \frac{\partial }{\partial t}p\left(t,y,x\right) & =-\sum_{w=x,y}\frac{\partial }{\partial_{w}}\left[\sum_{i,j}\left(i,j\right)\cdot C^{\left(i,j\right)}p\left(t,y,x\right)\right]+\frac{1}{2}\sum_{w,q=x,y}\frac{\partial }{\partial_{w}}\frac{\partial }{\partial_{q}}\left[ijC^{\left(i,j\right)}p\left(t,y,x\right)\right]=\\ \; & =-\sum_{w=x,y}\frac{\partial }{\partial_{w}}\left[B(x)p(t,y,x)\right]+\frac{1}{2}\sum_{q=x,y}\sum_{w=x,y}\frac{\partial }{\partial_{w}}\frac{\partial }{\partial_{q}}\left[Q(x)p(t,y,x)\right]\;, \end{array}$$

*where the above coefficients in Equation (Equation 23) are given explicitly by:*

$$\begin{array}{cc} B(x) & =\left(\begin{array}{c} x(x-1)b+xa\\ -2x(x-1)b-x(a+r) \end{array}\right)\;,\\ Q(x) & =\left(\begin{array}{cc} x(x-1)b+xa & -2x(x-1)b-xa\\ -2x(x-1)b-xa & 4x(x-1)b+x(a+r) \end{array}\right)\;. \end{array}$$


*The connection between Equations (Equation 14) and (Equation 23) can be made rigorous by introducing the new variables:*

$$\bar{x}:=\frac{x}{K}\;,\;\bar{y}:=\frac{y}{K}\;,$$

*into Equation (Equation 23), to obtain:*

(24)
$$\frac{\partial }{\partial t}p\left(t,\bar{y},\bar{x}\right)=-\sum_{w=\bar{y},\bar{x}}\frac{\partial }{\partial_{w}}\left[\tilde{B}\left(\bar{x}\right)p\left(t,\bar{y},\bar{x}\right)\right]+\frac{1}{2K}\sum_{w,q=\bar{y},\bar{x}}\frac{\partial }{\partial_{w}}\frac{\partial }{\partial_{q}}\left[\tilde{Q}\left(\bar{x}\right)p\left(t,\bar{y},\bar{x}\right)\right]\;.$$


*Performing a small-noise expansion [23,31,32], we can expand Equation (Equation 24) around $\epsilon :=\frac{1}{K}$, so that, using the new variables:*

$$\xi =\sqrt{K}\left(\bar{x}-\bar{x}\left(t\right)\right)\;,\;\upsilon =\sqrt{K}\left(\bar{y}-\bar{y}\left(t\right)\right)\;,$$

*we can see that the small noise expansion to the lowest order does coincide with the linear FP Equation (Equation 14).*

*The two expansions have different advantages and disadvantages. In fact, on the one side, considering the full expansion as described above, the nonlinearity of the system is preserved. However, on the other side, given the nonlinear diffusion term, the simulation of Equation (Equation 23) is more complicated. Further, since Equation (Equation 14) is linear, its solution can be computed analytically, showing that the process follows a Gaussian distribution.*


## 4. Results

The present Section reports the implementation of the results derived in Section 3. Figure 1 shows a comparison between the distribution of sublethal lesions (top row) and lethal lesions (bottom row) for the solution to the MME (Equation 2) (histogram) and the solution to the FPE (Equation 14) (yellow line). Different columns refer to different times: the left column reports time 0.5 [a.u.], the central column reports time 0.7 [a.u], and the right column reports time 0.9 [a.u.]. The solution of the FPE has been centered around the average values $\bar{x}$ and $\bar{y}$, whereas the MME (Equation 2) is solved via the *stochastic simulation algorithms* (SSA) [33] (Chapter 13). A deterministic initial value of $x_{0}=100$ and $y_{0}=0$ has been prescribed. Further, GSM${}^{2}\;$parameters have been chosen to be r=4, a=0.1, ${\tilde{b}}_{K}=0.01$; these parameters are in agreement with the parameters typically used [20]. A good agreement can be seen between the system size approximation and the original solution to the MME, particularly for lethal lesions. The approximation, as expected, shows a small discrepancy in the case of sublethal lesions at higher times, since the solution is closer to 0.

Figure 2 shows the time evolution for the moments Equations (Equation 11)–(Equation 16): in yellow is the solution to the average number of sublethal lesions $\bar{x}$, whereas in blue is the average number of lethal lesions $\bar{y}$. In black is depicted the covariance between lethal and sublethal lesions $c_{\xi \upsilon }$, in purple the variance of lethal lesions $c_{\xi \xi }$, and in red the variance of lethal lesions $c_{\upsilon \upsilon }$. Both the average and variance of lethal lesions converge to 0 for a long time. By contrast, the average and the variance of lethal lesions converge toward a strictly positive value, with the latter being strictly lower than the former. Additionally, the covariance is strictly negative and converges to 0 at long times.

Figure 3 shows a comparison of 10 path solutions of the average values (black), original GSM${}^{2}\;$formulation (yellow), and linear noise approximation (blue) for sublethal lesions (left panel) and lethal lesions (right panel). It can be seen how the approximation and the original GSM${}^{2}\;$formulation produce similar patterns, with the average values being in the middle of the stochastic solutions.

## 5. Discussion

In the present paper, we presented a linear noise approximation of a stochastic model for radiation-induced DNA damage repair and kinetics [10]. Such approximation is carried out by expanding around the system size so that that it holds true for a high number of particles, which can be approximated as a continuum. The fluctuations, for the number of particles sufficiently far from the origin, are predicted to be Gaussian-distributed. The importance of the result is twofold: (i) it allows for the fast computation and simulation of GSM${}^{2}\;$as certain, and (ii) it theoretically shows that the number of lethal lesions deviates from a Poisson distribution, as typically assumed in the vast majority of radiobiological models.

The results show a good agreement between the solution to the MME (Equation 2) and the linear noise approximation (Equation 14). This is particularly true when the number of lesions is far from the origin. In fact, in this situation, the description of GSM${}^{2}\;$as a continuum of lesions is not valid and discrepancies between the two representations emerge. This is, however, mitigated by equipping the linear FPE with a suitable boundary condition, preserving the positivity of the solution. Further, the main interest in the long-time distribution lies in the distribution of lethal lesions, which could allow characterizing several relevant biological endpoints such as cell survival and cell killing. This implies that such approximation can be effectively used in several concrete applications even if it is typically difficult to estimate the experimental range in which the approximation proposed is valid.

The numerical solutions to the moments Equations (Equation 11)–(Equation 16) confirm the theoretical analysis performed in Section 3. In particular, the variance of the lethal lesions is strictly lower than the average; this, together with the Gaussianity of the distribution, implies a divergence from the Poissonianity of the number of lethal lesions. This is one of the first theoretically grounded results showing that a model can predict lethal lesions with non-Poisson distribution. Additionally, the covariance is strictly negative, since an increase in the number of lethal lesions can only be caused by a decrease in the number of sublethal lesions.

Finally, it is worth remarking that this paper furthers the investigation and comparison of diverse existing radiobiology models, revealing the underlying commonalities and shared perspectives among these approaches. The present paper shows the connections of GSM${}^{2}\;$to two other models proposed in the literature. Firstly, the main equations of the MKM arise formally within the context of GSM${}^{2}\;$, with, however, an extremely relevant difference in the fluctuations around the average values. This has been already noted in previous research [10,19]. Secondly, the incorporation of a Gaussian distribution, previously employed in radiobiology studies [24,25], emerges as a deviation from a Poisson distribution. As a result, the proposed model establishes a remarkably insightful link between two seemingly different radiobiological models: the MKM and the Gaussian formulation of a multi-hit model. Recognizing the significance of this subject, future research will be dedicated to further exploring the interconnections among diverse radiobiological models.

## 6. Conclusions

The present research continues the investigation of how the stochastic nature of energy deposition affects DNA damage evolution and how this is, in turn, related to the overall probability distribution of the number of lethal and sublethal lesions. In [10], a *master equation* for the probability distribution of DNA damage has been derived. However, due to the non-linear terms, besides some cases such as the computation of the survival probability [18], its analytical solution is unfeasible. In the present work, we have shown how a proper expansion can be applied to the MME derived in [10]. Such expansion highlights, on one side, how the ${\mathrm{GSM}}^{2}$ is connected to the MKM and, on the other side, how non-Poissonian effects naturally emerge with no need for *ad hoc* corrections.

## Figures and Tables

**Figure 1 entropy-25-01322-f001:**
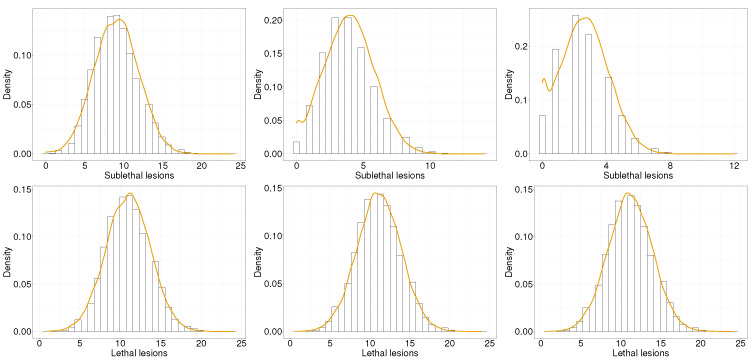
Sublethal lesion density evolution (**top row**) and lethal lesion density evolution (**bottom row**) as predicted by GSM${}^{2}\;$(histogram) and the linear noise approximation (yellow line). The left column reports time equal to 0.5 [a.u.], the central column reports time equal to 0.7 [a.u], and the right column reports time equal to 0.9 [a.u.].

**Figure 2 entropy-25-01322-f002:**
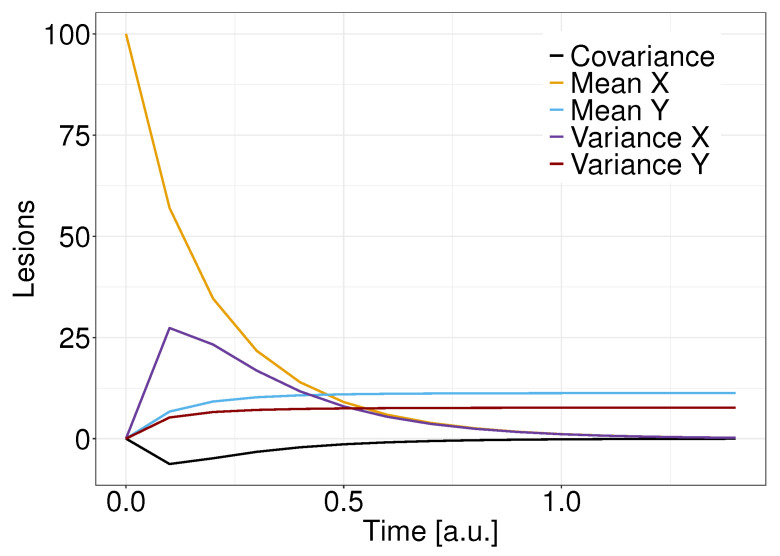
Time evolution for the average number and variance of sublethal lesions (yellow and purple), the average number and variance of lethal lesions (blue and red), and the covariance of lethal and sublethal lesions (black).

**Figure 3 entropy-25-01322-f003:**
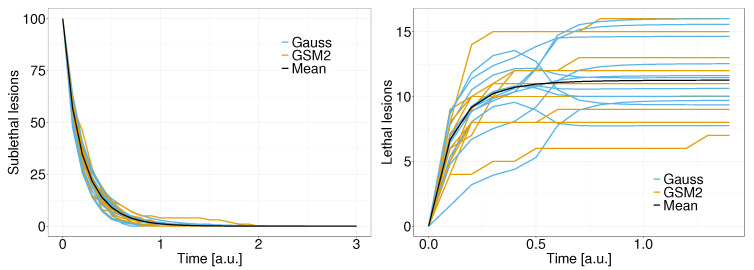
Comparison between path solutions to the GSM${}^{2}\;$(yellow), the linear noise approximation (blue), and the average value (black) of sublethal lesions (**left panel**) and lethal lesions (**right panel**).

## Data Availability

No new data have been created.

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
