# Peer review of "On the Emergence of the Deviation from a Poisson Law in Stochastic Mathematical Models for Radiation-Induced DNA Damage: A System Size Expansion"

_entropy, 2023, doi:10.3390/e25091322_

Round 1
Reviewer 1 Report
The questions addressed in the paper are in my opinion interesting, even if strongly based on previous work (also from the author). The improvements in the paper (compared with previous results) are clearly described. There is a satisfactory section dedicated to numerical simulations, confirming what previously described. The results obtained are sufficiently interesting to be published in Etropy.
Some very minor misprints here and there.
Author Response
I would like to thank the reviewer for her/his comments. The typos have been corrected. In the new version of the manuscript, changes are highlighted in blue.

Reviewer 2 Report
On the emergence of the deviation from a Poisson law in stochastic mathematical models for radiation-induced DNA damage: a system size expansion
by Francesco G. Cordoni
General comment:
This paper addresses the relationship between stochastic and deterministic mathematical models for radiation-induced DNA damage repair. An asymptotic expansion for the GSM2 master equation computing both the macroscopic limit and the fluctuation around such macroscopic limit is derived. The results are simulated by choosing the numerical values of the parameters and the initial conditions. The manuscript proposes an innovative technique, which can capture key aspects in a specific area.
In this work, the author summarizes his previous research, unfortunately, the model is linear, the growth is exponential, the derived density is only Gaussian. Real processes are characterized by asymmetry, and the evolution is usually sigmoidal.
The article could be valuable with these corrections:
1. Line 141: Since p() is a transition probability density function, this should specify the initial state and time.
2. Line 142: Undefined constants: r, a, b.
3. Line 143: Undefined operators: E?, Ɛ?.
4. Line 150: Are these univariate independent distributions?
5. Line 183 (and below): ξ and v depend on t, so they are not random variables - they are probably random processes.
6. Line 186: …standard differential equation… Unclear.
7. Lines 232-233: Correct y0 and resp.!
8. Line 234: Add reference.
9. Lines 236-238: You should be talking about the Gaussianity of a two-dimensional random process.
10. Lines 265-267: Add an initial condition that could additionally have a bivariate normal distribution.
With this in mind, I propose a major revision of this article.
The article is written in acceptable language.
Author Response
I thank the reviewer for her/his comment.
Detailed answers are the following. In the revised manuscript, changes are highlighted in blue.
I added a paragraph at line 149 regarding a possible logistic parameter. Also, I added a Remark below line 309 with some comments on a possible more general expansion where the derived equation is nonlinear.
- I added a precise formulation including the initial state.
- I added a line at line 140 introducing the rates.
- the operators are defined in line 146.
- the distribution can be either the product of two univariate independent distributions or also a bidimensional distribution. The second case is obviously more general, but in practical application, the resulting distribution is very close to the product of two independent univariate distributions, so this latter case is typically assumed.
- it has been corrected.
- I removed the word "standard".
- it has been corrected.
- I added a comment on this.
- I postponed the Remark at line 296 including the initial condition.

Round 2
Reviewer 2 Report
The authors answered all the questions that arose. Paper can be accepted.
The article is written in acceptable language.